# The Role of the Immune System in the Development of Endometriosis

**DOI:** 10.3390/cells11132028

**Published:** 2022-06-25

**Authors:** Monika Abramiuk, Ewelina Grywalska, Paulina Małkowska, Olga Sierawska, Rafał Hrynkiewicz, Paulina Niedźwiedzka-Rystwej

**Affiliations:** 1Department of Gynecological Oncology and Gynecology, Medical University of Lublin, 20-081 Lublin, Poland; monika.abramiuk@gmail.com; 2Department of Experimental Immunology, Medical University of Lublin, 20-093 Lublin, Poland; ewelina.grywalska@gmail.com; 3Institute of Biology, University of Szczecin, 71-412 Szczecin, Poland; paulina.malkowska@phd.usz.edu.pl (P.M.); olga.sierawska@phd.usz.edu.pl (O.S.); rafal.hrynkiewicz@usz.edu.pl (R.H.); 4Doctoral School, University of Szczecin, 71-412 Szczecin, Poland

**Keywords:** endometriosis, immune system, cytokine, defensins

## Abstract

Endometriosis is a chronic disease that affects about 10% of women of reproductive age. It can contribute to pelvic pain, infertility or other conditions such as asthma, cardiovascular disease, breast or ovarian cancer. Research has shown that one of the conditions for the development of endometrial lesions is the dysfunction of the immune system. It appears that immune cells, such as neutrophils, macrophages, NK cells and dendritic cells, may play a specific role in the angiogenesis, growth and invasion of endometriosis cells. Immune cells secrete cytokines and defensins that also affect the endometriosis environment. This review discusses the various components of the immune system that are involved in the formation of endometrial lesions in women.

## 1. Introduction

Endometriosis is a common chronic disease of women of reproductive age. It affects approximately 10% of women [1,2]. The classic definition of endometriosis refers to the presence of endometrium-like tissue outside the uterine cavity; however, there is considerable heterogeneity among patients, both in terms of disease phenotype and associated symptom severity. Although chronic inflammation and unusually high estrogen levels are well-known features of endometriosis, the exact etiology of the disease remains largely elusive [3]. This can be attributed to the complex and multifactorial nature of the disease, in which the involvement of genetic, hormonal, environmental, and immunological factors has previously been identified [4]. One of the major challenges of modern gynecology is to understand the pathophysiology of endometriosis and attempt to prevent it. To date, the most widely accepted theory of endometrial lesion formation is that during menstruation, endometrial cells and tissue fragments retract through the fallopian tubes and attach to pelvic structures, causing an inflammatory response, fibrosis, and pain [5]. Cytokines are responsible for the inflammatory response and its regulation. The activation of immune cells triggers signaling pathways that cause the release of inflammatory cytokines, which then contribute to the accumulation of multiple cell types at the site of inflammation [1]. During research, it has been shown that one of the conditions for the development of endometrial lesions is the dysfunction of the immune system, which affects the expression of particular cytokines [6,7].

At present, there is no unified classification system for endometriosis. Four standard systems have emerged (the revised American Society for Reproductive Medicine (rASRM) classification, ENZIAN classification, endometriosis fertility index (EFI), and American Association of Gynecological Laparoscopists (AAGL) classification), but their utility remains controversial [8]. The first classification that has been accepted worldwide and is widely used is the rASRM. It is based on determining the severity of endometriosis using a cumulative score. The grading system is divided into four stages: I minimal (1–5 points), II mild (6–15 points), III moderate (16–40 points) and IV severe (>40 points). Endometriosis lesions were classified as superficial and deep lesions. The size of deep ovarian endometriosis > 3 cm scored 20 points, and dense ovarian adhesion and dense fallopian tube obstruction were classified as 16 points. In addition, a single finding of complete blind overgrowth scored 40 points and was classified as severe disease [9]. The ENZIAN score, like the rASRM classification, is determined by the extent of endometriosis at surgery. It is based on dividing the retroperitoneal structures into three compartments. The posterior uterus is divided into compartment A consisting of the rectovaginal septum and vagina, compartment B consisting of the uterosacral ligament and pelvic walls, and compartment C consisting of the sigmoid colon and rectum. The severity of the lesion is set to invasiveness: grade I < 1 cm, grade II—invasiveness between 1 and 3 cm, and grade 3—invasiveness > 3 cm. The presence of an endometriosis tumor is indicated by the prefix “E”. The number following the prefix indicates the size of the lesion, and after the number a lowercase English letter indicates the affected compartment. Two lowercase English letters indicate bilateral disease. The invasion of endometriosis into other organs in the pelvic cavity and into distant organs is expressed as follows: “FA” is defined as adenomyosis, “FB” as bladder involvement, “FU” as internal ureteral involvement, “FO” as involvement of other locations, and “FI” as intestinal involvement [8]. The EFI system aims to predict pregnancy rates in patients with surgically documented endometriosis who have not attempted in vitro fertilization (IVF). The functional assessment indicates whether the embryo is well implanted in the uterus, whether the uterus can provide an early environment for the embryo, or whether the fallopian tubes are able to retrieve the egg well. It is determined by the surgeon and ranges from 0 to 4 points: 0—absent or nonfunctional, 1—severe dysfunction, 2—moderate dysfunction, 3—mild dysfunction and 4—normal. The rASRM score is also included. The EFI score is calculated by summing the historical and surgical scores and ranges from 0 to 10 points, with 10 indicating the best prognosis and 0 the worst prognosis [10]. The final classification, AAGL, is based on assigning a score from 0 to 10 based on the importance of each disease site to the outcomes of pain, infertility, and surgical difficulty. In addition, surgical difficulty was divided into four levels: level 1—excision or desiccation of superficial implants and simple thin nonvascular adhesions; level 2—removal of ovarian endometrial tumors, excision of a blind bowel, deep endometriosis not involving the vagina, bladder (not requiring sutures), bowel or ureter, dense adhesions not involving the bowel and/or ureter; level 3—dense adhesions involving bowel and/or ureter; bladder surgery requiring sutures, ureterolysis, bowel surgery without resection (shaving); level 4—bowel resection with end-to-end anastomosis; reimplantation or ureteral anastomosis [11].

The study literature confirms the involvement of immune cells in the pathogenesis of endometriosis. Neutrophils and peritoneal macrophages produce biochemical factors that aid in angiogenesis and endometriotic cell growth and invasion. Peritoneal macrophages and NK (natural killer) cells in endometriosis have a limited ability to eliminate endometrial cells in the peritoneal cavity. The imbalance between Th1/Th2 cells leads to abnormal cytokine secretion and inflammation, which continues to induce lesion progression [12]. A promising factor involved in the pathogenesis of endometriosis seems to be immune checkpoint inhibitors, as indicated in very recent literature [13]. It is still uncertain whether disorders of the immune system initiate endometriosis; however, the course of the disease can be affected by modulating its actions [14].

According to Dmowski and Braun [15], abnormal, genetically polymorphic endometrial cells may respond to local endometrial signals with proliferation rather than apoptosis. These cells, when abnormally displaced into the peritoneal cavity, may not only avoid peritoneal destruction but also use the peritoneal environment to their advantage and continue to proliferate in a clonal fashion while normal cells in the same individual are eliminated. The products secreted by these cells provide an additional source of cellular invasion and disease progression, and induce a local inflammatory response. This causes chemotaxis of leukocytes into the peritoneal cavity, their activation and secretion of pluripotent cytokines. In addition, these mobilized immune cells may have a reduced ability to eliminate abnormally distributed cells, and the products they secrete may stimulate rather than inhibit ectopic cell proliferation and disease progression. Functional abnormalities in immune cells in endometriosis may result from exposure to the same environmental toxins that cause alterations in B, NK, and T cell function in women with endometriosis.

## 2. The Role of Immunocompetent Cells in the Development of Endometriosis

The innate immune system cell populations that are involved in the pathophysiology of endometriosis include neutrophils, macrophages, NK cells, and dendritic cells (Figure 1).

In the context of endometriosis, increased percentages of neutrophils have been observed in patients with endometriosis compared to women without the disease [16,17]. This is probably the result of increased concentrations of chemotactic factors such as IL-8 present in plasma and peritoneal fluid, epithelial neutrophil-activating peptide (ENA-78), and human neutrophil peptide (HNP1-3) in the local endometriosis environment [18]. 

Interestingly, estrogens have been shown to influence the accumulation and modification of neutrophil function [19]. However, it has not been proven whether this property may have a role in the pathogenesis of endometriosis. In line with the emerging concept of neutrophil heterogeneity, attempts should also be made to determine whether neutrophil phenotype and function are altered in the microenvironment of endometriosis lesions [17].

In an animal model of endometriosis, neutrophil infiltration into ectopic uterine tissue was dominant in the early stages of the disease [20,21]. The role of these cells is largely to generate inflammation, particularly through the secretion of IL-17A, which enhances neutrophil migration, or other pro-inflammatory factors: IL-8, VEGF, CXCL10 [22].

Macrophages also play a role in enhancing inflammation and following with neutrophil recruitment through the release of chemokines. These two populations are observed in increased numbers within endometrial lesions [23]. However, the phagocyte function of peripheral monocytes and neutrophil granulocytes seem to be decreased and its activity is affected by the presence or removal of endometriotic lesions in women with endometriosis [24].

Macrophages play a key role in the development of endometrial lesions and the concomitant inflammation. However, the overall number of monocytes in the peripheral blood of endometriosis patients did not alter [25,26]. In the peritoneal fluid, they are the most abundant cells found in healthy individuals; therefore, it is not surprising that their elevated levels have been observed in the local endometriosis environment. In their presence, the development of endometrial implants is more intense. 

A significant increase in the number of macrophages has also been shown in the eutopic endometrium in women with endometriosis [27,28]. By using an animal model for endometriosis, Haber et al. demonstrated that the inhibition of peritoneal macrophage function impaired the development and growth of endometrial implants [29]. Further studies have shown that macrophages in endometriosis patients are not fully functional. The reduced expression of CD3 (cluster of differentiation 3) and annexin A2 is represented by reduced phagocytic capacity [22]. Furthermore, the production of inflammatory mediators by macrophages favors the implantation and proliferation of endometrial cells, resulting in the development of endometrial lesions [30]. 

Macrophages were identified to co-occur in increased numbers with nerve fibres in peritoneal endometrial lesions [31]. It has been hypothesized that macrophages and nerve fibres interact to promote pain symptoms associated with endometriosis. It is worth noting that Greaves et al. recently discovered that estradiol is a key mediator of the interactions between macrophages and nerve tissue in peritoneal endometriosis. This thesis was confirmed both in vitro and in a mouse model [17,32].

NK cells, predominantly characterized by the following surface markers: CD56^dim^CD16+ and CD56^bright^CD16−, are the link between two immune response types—innate and adaptive. The involvement of NK cells in the pathogenesis of endometriosis was first described by Oosterlynck et al., who stated that the cytotoxic activity of NK cells was lower against ectopic endometrial cells. This relationship correlated with advanced stages of the disease [20,33,34]. Since then, it has been a well-established observation and, although more pronounced for NK cells in the peritoneal fluid, it is also observed for NK cells in peripheral blood [20,35]. The mechanism of NK cell function inhibition in endometriosis was not elucidated. The abnormal expression of various activating and inhibitory receptors on their surface compared to the population of these cells in healthy women was described [36,37,38,39]. There was reported an increased expression of KIR (killer cell immunoglobulin-like receptors) on the surface of peritumoral NK cells, which may explain this [40]. It is also possible that this abnormal NK cell function is a consequence of chronic disease-induced inflammation [41,42]. NK cells contribute to the balance of immune tolerance by eliminating cells that present autoantigens; therefore, their reduced activity in endometriosis may explain the more frequent autoimmunity observed in the disease [43].

Recent years have brought progress in identifying factors that may impair NK cell function in endometriosis, as researchers seek to clarify the role of specific cytokines and chemokines in the pathophysiology of the disease. Both IL-6 and transforming growth factor (TGF-β) in the peritoneal fluid of women with endometriosis have been shown to reduce the cytolytic activity of NK cells [44,45]. IL-6 has also been shown to affect the number of cytolytic components of NK cells by modulating Src homology region 2-containing protein tyrosine phosphatase-2 [45]. Additionally, IL-15, which is highly expressed on endometrial stromal cells in the ectopic location, has been demonstrated to inhibit NK cell function in vitro [46]. By imitating the local microenvironment in endometriosis, Yang et al. cultured stromal cells from endometrial implants from patients with endometriosis with both monocyte-derived macrophages and NK cells. As a result, it was found that the interaction of endometrial cells with macrophages reduced NK cell cytotoxicity, probably through increased release of IL-10 and TGF-β [47].

Among the numerous studies on the cellular response in the pathogenesis of endometriosis, several aspects are worth noting. Due to the increased concentrations of cytokines typical of Th2 lymphocytes in the plasma and the peritoneal fluid of patients with endometriosis, the disease was characterized as polarized towards this particular cellular response [48,49]. However, the typical Th2 response, which is physiologically associated with healing and fibrosis, has not been fully observed in the context of endometriosis [50]. The interpretation of studies is complicated by a relative predominance of Th1 lymphocytes observed in peripheral blood. Studies evaluating T cells in patients with endometriosis revealed a higher CD4/CD8 ratio and an increased concentration of each of these cell types in the peritoneal fluid of patients [51]. Endometrial lesions showed higher concentrations of T cells compared to those in the eutopic endometrium, but with a similar CD4/CD8 ratio. No changes in peripheral blood were observed [52,53]. Different results were presented in our study, where a decrease in the number of CD8+ T lymphocytes in peripheral blood was observed, with no significant deviation in the total number of lymphocytes. Consequently, the CD4/CD8 ratio in blood was higher in the group of women with endometriosis compared to the control group. At the same time, it has been shown that the percentage of Th17 cells in the peritoneal fluid of patients is increased and related to the stage of the disease [54]. Higher levels of IL-17 have also been shown to occur in the endometriotic environment. This interleukin may progress endometriosis by stimulating the production of cytokines that induce angiogenesis and inflammation [55,56].

Among the lymphocytes involved in the cellular response, regulatory T cells play an important role (Figure 2). In principle, they have a regulatory function and are involved in the development of immunological tolerance, but their role in the development of endometriosis remains unclear. Authors almost unanimously indicate an increased number of regulatory T cells in the peritoneal fluid of patients, suggesting that this is associated with local immunosuppression, hindering the elimination of ectopic endometrial cells. Hanada et al., who demonstrated a higher percentage of Treg in the peritoneal fluid of patients, at the same time found no changes in peripheral blood in this respect compared to controls [53]. Similar results were presented by Olkowska-Truchanowicz et al., who evaluated the percentages of CD4+CD25+Foxp3+ T lymphocytes in peripheral blood, which were also consistent with the results of Takamura et al. [57,58]. Our study also revealed no differences in the percentage of regulatory T cells with the CD4+CD25+Foxp3+ phenotype in peripheral blood, suggesting that their role is limited to the local effect.

### Humoral Immune Response

Endometriosis has repeatedly been viewed as an autoimmune disorder. Various reports confirmed the presence of anti-endometriosis antibodies in the serum and peritoneal fluid of patients diagnosed with the disease [59,60]. Autoantibodies are generally believed to contribute to the development of endometriosis by stimulating the immune system and perpetuating inflammation; however, there is no conclusive evidence to support the thesis. The fact that autoantibodies are present in patients with endometriosis is further complicated by comorbidities. A clear relationship is emphasized between endometriosis and the presence of other immunological and autoimmune diseases, such as rheumatoid arthritis, psoriasis and allergies [31,61]. Due to the presence of autoantibodies, the role of B cells in the pathophysiology of endometriosis was also investigated. In the published meta-analysis, most studies suggest an association with an increased percentage or activation of B lymphocytes in patients with endometriosis. In 7 of the 22 included studies, no differences in B cell numbers were observed between patients and controls [32]. Apart from antibodies, B cells also produce cytokines such as IL-6 (granulocyte-macrophage colony-stimulating factor) and IL-17, which have been shown to modulate immune cells (e.g., CD4+ T cells) and perpetuate chronic inflammation [34]. These cytokines are also associated with endometriosis [18,33,62]. B cells may therefore participate in systemic and local cytokine production in endometriosis and cause inflammation in their own microenvironment. Further study is thus warranted to fully understand the role of B cells and their interaction with other immune cells in the endometriosis microenvironment.

## 3. The Role of Selected Cytokines in the Pathogenesis of Endometriosis

The relationship of endometriosis—as a local, chronic inflammatory disease—with excessive cytokine secretion has been repeatedly discussed. As chemotactic factors, cytokines are involved in the recruitment of macrophages and T cells, mediating the inflammatory response correlated with the disease (Table 1). Researchers observed altered cytokine expression in serum or peritoneal fluid in the presence of endometriosis in different locations [63,64].

### 3.1. Interferon Gamma (IFN-γ)

The main biological activities resulting from interferon (IFN) signalling are antiviral, antiproliferative, antiangiogenic, and inhibitory activities to antigen presentation. IFN also affects the regulation of genes and pro- and anti-apoptotic proteins [65]. The current state of cells and complex feedback mechanisms determine the overall outcome of the IFN-mediated response, e.g., survival vs. apoptosis. IFN-γ is mainly produced by activated T or NK cells, and its role is to modulate the cellular immune response, which, among other things, contributes to macrophage activation and T cell development [66]. 

INF-γ is another important cytokine that interacts with IL-2 in regulating the balance between Th1 and Th2 cells. By examining peripheral blood and peritoneal fluid, Hsu et al. demonstrated reduced levels of INF-γ in both body fluids. However, Podgaec et al. found a higher expression of IFN-γ in the local environment of lesions, i.e., peritoneal fluid in patients with endometriosis compared to controls, with no statistically significant differences between its systemic levels, i.e., in peripheral blood [12]. Similar results regarding the microenvironment of endometriosis were obtained by showing that IFN-γ mRNA expression is significantly higher in ectopic tissue compared to eutopic endometrium [67,68,69]. Interestingly, it was noted that IFN-γ does not affect the growth and apoptosis of cells from ovarian-derived ectopic endometrial implants [70]. This may indicate that endometrial cells become resistant to apoptotic signals when they enter the peritoneal cavity through retrograde menstruation [70].

### 3.2. Interleukin 1 (IL-1)

IL-1 is a pro-inflammatory family of cytokines that are secreted by activated peritoneal macrophages into the peritoneal fluid [71]. IL-1 has several functions in the organism, the most important of which is the regulation of immune and inflammatory responses [72]. IL-1 initiates an inflammatory response cascade primarily through cytokines (IL-6, IL-8), B cells, and antibodies, and the secretion of specific matrix metalloproteinases (MMPs) and prostaglandins [20,71,73]. Together with other substances, IL-1 can induce the growth and differentiation of many cell types and is involved in angiogenesis [71]. IL-1 levels are elevated in women with endometriosis [71,74,75]. This is true for both IL-1α and IL-1β levels, although their effects on endometrial cells differ [74]. Changes in immune cell function in the peritoneal fluid and cytokine production, including IL-1, have been reported to be one cause of infertility [76]. Increased levels of IL-1α are observed in the peritoneal fluid of women with endometriosis. In the advanced stage of the disease, elevated levels of this cytokine are observed not only in the peritoneal fluid but also in the serum [74]. Furthermore, compared to the eutopic endometrium, the endometriotic lesions were characterized by more pronounced immunohistochemical staining for IL-1α [73]. IL-1β plays an important role in the formation of new blood vessels in tissues surrounding endometriotic lesions by producing vascular endothelial growth factor (VEGF) and IL-6 [71,74]. It also affects the proliferation of endometrial-derived cells [74] and regulates the expression of ICAM-1 on the cell surface [71]. In general, the main function of IL-1β is to regulate the inflammatory response [77]. However, unrestricted IL-1 secretion could lead to tissue damage and chronic inflammation, but self-regulation of the IL-1 family has been shown [74].

### 3.3. Interleukin 2 (IL-2)

Interleukin 2 (IL-2), together with the specific interleukin 2-receptor (IL-2R), is a cytokine that has been studied due to its primary activity, i.e., participation in cytotoxic cellular response. In addition, it stimulates the proliferation and differentiation of B and T cells, the activation and proliferation of nonspecific cytolytic effector cells, including NK cells and lymphokine-activated killers (LAKs), and is involved in the activation of monocytes and macrophages [78]. 

The role of IL-2 in the pathogenesis of endometriosis is not clearly defined. This is evidenced by the differing results presented by other researchers. Hsu et al. found decreased concentrations of IL-2 in both peripheral blood and peritoneal fluid, more pronounced in advanced stages of the disease. At the same time, they observed no differences in IL-2 and IL-10 mRNA levels between the study group and the control group [79]. Similar results were presented by Hernandez-Guerrero et al. showing lower intracellular IL-2 synthesis in peritoneal fluid and peripheral blood in patients [80]. The same findings were presented by Ho et al. [35]. Gogacz et al., on the other hand, reported an increase in IL-2 levels in the peritoneal fluid in patients with endometriosis and coexisting infertility [54]. In the case of infertile patients with endometriosis, similar results were obtained by Chinese researchers [81].

### 3.4. Interleukin 6 (IL-6)

Intensely and periodically produced in response to infection and tissue damage, interleukin 6 (IL-6) induces host defence by stimulating acute phase responses, hematopoiesis, and immune responses. Its expression is strictly controlled by transcriptional and post-transcriptional, non-regulated mechanisms. However, under pathological conditions, there may be a continuous synthesis of IL-6, which plays a role, inter alia, in chronic inflammation and autoimmunity [82,83]. IL-6 exerts much bioactivity through its receptor (IL-6R). Membrane-bound receptor (mIL-6R) and soluble receptor (sIL-6R) are two forms of IL-6R.

According to the literature examples, IL-6 promotes the occurrence and development of ectopic endometrial foci by interfering with the cellular response [84,85,86,87,88,89]. The predominant cells secreting IL-6 in the peritoneal fluid are macrophages [1]. Many studies showed that activated macrophages in women with endometriosis are present in significantly higher numbers and secrete cytokines, including IL-6, more intensively [45,90,91]. IL-6 secreted by activated macrophages has a pleiotropic effect, and one of its effects is an increase in haptoglobin, which “protects” endometrial implants from immune surveillance by reducing phagocytosis. The positive feedback loop created in this way facilitates the survival of the ectopic endometrium and promotes the development of endometriosis foci [92]. It is also worth noting that elevated IL-6 levels inhibit NK cell activity in the peritoneal fluid of patients with endometriosis [45].

The altered local expression of IL-6 may be important in the pathogenesis of endometriosis. Many studies highlight the presence of increased interleukin 6 levels in affected patients [45,93]. Li et al. showed that the increase in its concentration in the peritoneal fluid is more significant compared to the inconclusive serum results obtained. At the same time, they emphasized that the change in concentration was not related to the stage of the disease [94].

Studies revealed that IL-6 and IL-6R act as growth regulatory signals for human endometrial stromal cells. Endometrial stromal cells can become resistant to IL-6 by reducing IL-6R expression [95,96].

### 3.5. Interleukin 8 (IL-8) and Tumor Necrosis Factor-α (TNF-α)

IL-8 and TNF-α were among the cytokines whose higher expression levels were noted in the peritoneal fluid of women with endometriosis [97,98,99]. IL-8 is a chemotactic factor that attracts cells such as neutrophils, basophils and T lymphocytes to the site of inflammation. It is released from monocytes, macrophages, neutrophils, among others, in response to inflammation and is involved in neutrophil activation [100]. TNF-α is produced by macrophages/monocytes in response to acute inflammation. It is responsible for a number of signaling pathways leading to necrosis or apoptosis [101].

During the study, we found that increased levels of IL-8 in the peritoneal fluid of women with endometriosis contributed to higher proliferation of ovarian endometrioma-derived stromal cells. Based on this, it was concluded that IL-8 may contribute to the development of endometriosis [97]. In addition, TNF-α was found to contribute to increased gene and protein expression of IL-8 by also stimulating endometriosis-derived stromal cell proliferation [99]. These results allow us to conclude that IL-8 and TNF-α are factors that facilitate the adhesion of endometrial cells to the peritoneum and may contribute to the progression of the disease [1].

### 3.6. Interleukin 10 (IL-10) and Interleukin (IL-4)

Interleukin 10 (IL-10) is a potent anti-inflammatory cytokine whose deficiency or abnormal expression may increase the inflammatory response or lead to the development of a number of autoimmune diseases [102,103,104]. Initially, IL-10 production was mainly attributed to Th2 lymphocytes, but further studies have demonstrated that B cells, T cells, macrophages, as well as cells not directly involved in the cellular response, such as keratinocytes, and tumor cells can be sources of IL-10 [105]. In general, the term “anti-inflammatory effect” includes mechanisms such as inhibition of the release of pro-inflammatory mediators, counteracting phagocytosis, reducing antigen presentation while enhancing T cell regulatory functions and the tolerance environment [106,107]. Additionally, IL-10 may increase the activation and the proliferation of certain types of immune cells, including mast cells, CD8+ T cells, NK cells, and B cells. The molecular mechanisms and functional consequences of such IL-10 activity remain unclear [108,109,110].

Interleukin 4 (IL-4) is a cytokine with pleiotropic activity [111]. It is involved in immunoglobulin class switching [112], increases the expression of MHC class II molecules in B cells [113], enhances CD23 expression [114], increases IL-4 receptor expression [36], and influences the longer survival of T and B cells [37]. However, its main role is to participate in the differentiation of naive T cells after their stimulation with antigen. As a result of this process, activated T cells are able to produce helper T cells polarized towards a Th2 cellular response [38,39].

In the context of endometriosis, it has been repeatedly shown that the disease development is accompanied by the activation of Th2-type immune responses [12,79,115]. IL-4 and IL-10 are antagonists of the synthesis of inflammatory cytokines: IFN-γ, IL-2, IL-3, TNF-α, and GM-CSF [116]. The above cytokines, by promoting a Th2-type response, inhibit the cytotoxic response, whose involvement in the removal of ectopic endometrium has been postulated [117]. IL-4 in ectopic endometrial implants stimulates eotaxin secretion, which results in increased angiogenesis and lesion progression [75]. At the same time, its anti-inflammatory activity can inhibit the formation of adhesions in the peritoneal cavity [79]. It has been shown that cells capable of producing IL-4 are present in ectopic endometrial tissues and that IL-4 has a stimulatory potential against endometrial stromal cells [118]. 

Ho et al. found higher levels of IL-10 in the peritoneal fluid of women with endometriosis [119]. An association of elevated IL-10 levels with reduced NK cell cytotoxicity in the endometriosis environment was also presented [120]. In another study, Suen et al. discovered higher IL-10 levels also in peripheral blood plasma in patients with endometriosis compared to both healthy women and women with other gynecological conditions [121]. On the other hand, Ahn et al. observed that dysregulation of the local concentration of this cytokine facilitates the implantation of endometrial fragments into the peritoneal cavity [122]. Studies on a mouse model of surgically induced endometriosis allowed the authors to conclude that the growth of ectopic endometrium is promoted with a corresponding change in IL-10 concentration [121]. 

The presence of higher values of IL-4 and IL-10 in the peritoneal fluid confirms that the development of endometriosis is accompanied by the activation of a Th2-type immune response at the local level. However, the finding of a higher systemic concentration of Th1-related cytokines (IFN-γ, IL-2) with an increase in IL-10 and a decrease in IL-4 in peripheral blood may be a manifestation of a compensatory effect that antagonises inflammation. Podgaec et al. obtained similar, ambiguous results in their study, suggesting that both types of immune response are involved in disease development [12].

### 3.7. Transforming Growth Factor-β (TGF-β)

Studies in women with endometriosis have shown increased TGF-β activity in peritoneal fluid [123]. It is a highly pleiotropic cytokine that is involved in many processes including angiogenesis and immunoregulation. Its isoform, TGF-β1, has anti-inflammatory functions; however, TGF-β contributes to the differentiation of both regulatory T cells and Th17 inflammatory cells. It may be overproduced in some autoimmune diseases [124]. Researchers suggest that TGF-β in endometriosis may be responsible for inhibiting NK cell activity in the peritoneal fluid [123]. This theory is consistent with the assumption of retrograde menstruation, which posits that impaired NK cell function is the driver of endometrial cell survival and implantation in pelvic structures. Furthermore, decreased NK cell activity in the peritoneal fluid of women diagnosed with endometriosis also supports these assumptions [125,126,127].

It was found that the interaction of endometrial cells with macrophages reduced NK cell cytotoxicity [47]. Reduced NK cell cytotoxicity in endometriosis potentially contributes to a defective immune surveillance, but comprehensive analyses of the expression of activating and inhibitory receptors on NK cells are needed. Attempts should also be made to determine whether NK cells present in tissues in the endometrial microenvironment originate from the endometrium or peripheral blood, as these cell populations are likely to have different functions in the pathophysiology of endometriosis.

### 3.8. Vascular Endothelial Growth Factor (VEGF)

It is thought that angiogenesis, the process of creating new capillaries, may be involved in the pathogenesis of endometriosis. It is hypothesized that endometrial lesions may grow with the growth of new capillaries [1]. These assumptions are based on studies that have shown that tumor growth is dependent on angiogenesis [128].

Vascular endothelial growth factor (VEGF) is a potent heparin-binding angiogenic factor [129,130]. Because endometriosis is characterized by high vascularization within and around the ectopic tissue, it has been speculated that levels of the potent angiogenic growth factor VEGF in the peritoneal fluid may have strong clinical significance. Studies have shown that women diagnosed with endometriosis have higher levels of VEGF in the peritoneal fluid than healthy women. Furthermore, it was shown that the main source of VEGF is peritoneal fluid macrophages and that anti-VEGF antibodies abolished the increased proliferation of endothelial cells that were induced by the medium of macrophages isolated from the peritoneal cavity of women with endometriosis. On this basis, we are able to conclude that VEGF expression in endometriosis is regulated by estradiol and progesterone [131]. 

## 4. The Role of Defensins in the Pathogenesis of Endometriosis

Defensins belong to the family of antimicrobial peptides. In humans, there are two classes of defensins, α, and β. Among the α class, there are five defensins, four human neutrophil peptides (HNP1-4), and human defensin 5 (HD5). In the β class, there are four defensins named human beta-defensins (HBD1-4) [132].

Das et al. [132] showed that there was no significant increase in HNP1-3 expression in follicular fluid collected from women with infertility due to endometriosis compared to women with infertility due to lack of ovulation or ovarian factors [132]. Furthermore, another study by Das et al. found no significant changes in HNP1-3 levels in follicular fluid collected from women with endometriosis-induced infertility compared to the control group (male factor infertility group) [133]. These results may suggest that despite the presence of inflammatory processes in association with endometriosis, defensins are not affected in follicular fluid. In contrast, increased levels of HNP1-3 levels were observed in the peritoneal fluid of patients with endometriosis, compared to healthy women, and these levels were associated with disease progression [16].

β-defensin HBD2 is expressed in normal endometrial tissues and is thought to play an important role in maintaining the balance of reproductive mucosal defense mechanisms [134]. Chen et al. [134] reported an increase in HBD2 gene expression levels in tissues collected from patients with ectopic endometriosis compared to women with eutopic endometriosis and a control group [134]. Differences in HBD2 levels were also shown between the stages of ectopic endometriosis, where its levels were significantly higher in stages I-II than in stages III-IV. The main source of HBD2 expression was the cytoplasm of glandular epithelial cells in endometrial tissues, as well as the cell membrane and intercellular junctions [134]. The mRNA expression of defensins in the endometrium has been shown to vary with the phase of the menstrual cycle. HBD1 has maximum expression during the middle luteal (secretory) phase (where embryo implantation occurs), HBD2 during the menstrual phase, HBD3 during the early and late luteal (secretory) phases, and HBD4 during the proliferative phase [132]. Samples were taken during proliferative and secretory phases, so the cycle phase had no effect on HBD2 expression.

Increased levels of HNP1-3 in the peritoneal fluid of patients with endometriosis likely originate from infiltrating neutrophils. There is increasing attention being paid to the role of neutrophils in the pathogenesis of endometriosis. The release of large amounts of HNP1-3 may influence the pathogenesis of endometriosis and exacerbate its symptoms. HNP1-3 may be partly responsible for inducing local immune responses in the pelvic cavity of patients with endometriosis by recruiting T lymphocytes [16]. The role of human defensins in the pathogenesis of endometriosis still needs further investigation. It remains to be determined whether elevated levels of certain defensins are a causative factor in the development of the disease or a consequence of the disease.

## 5. The Role of Inhibitory Immune Checkpoints in the Pathogenesis of Endometriosis

The existing studies provide a lot of information on the main mechanisms involved in the pathogenesis of the disease, but many questions remain to be answered. The type of primary and secondary immune response cells in the microenvironment of endometriosis, as well as their relationship to the different stages, phenotypes and symptoms of the disease, are still not established. An important issue remains to assess how the resulting defects in immune “surveillance” in the altered microenvironment of endometriosis affect other autologous tissues.

### 5.1. Programmed Death Receptor 1 (PD-1) and Its Ligands

Programmed death receptor 1 (PD-1) is classified—together with the CD28 molecule, cytotoxic T-lymphocyte antigen-4 (CTLA-4), inducible costimulator (ICOS), and B- and T-lymphocyte attenuator (BTLA, CD272), B7-H3 (B7RP-2, CD276) and B7-H4 (B7x, B7S1)—in the group of CD28-B7 signalling receptors. It acts as a negative immunoregulator, inducing the cell cycle blockade [135,136]. This 288-amino-acid membrane receptor with a molecular weight of 50–55 kDa consists of three parts: the extracellular domain, the transmembrane domain (similar to the variable region of the immunoglobulin chain) and the cytoplasmic domain, in which there are the so-called immunoreceptor tyrosine-based inhibitory motif (ITIM) and immunoreceptor tyrosine-based switch motif (ITSM) [137,138,139,140]. Of these, ITSM plays an important role in PD-1 activity by interacting with the protein tyrosine phosphatases SHP-1 and SHP-2 [138,141].

The activation of lymphocytes by antigen stimulation leads to the increased expression of the PD-1 molecule on the lymphocyte surface [142]. The CTLA-4 and PD-1-mediated activation of signalling pathways results in the inhibition of T cell activity, while pathways including CD28 and ICOS affect cell activation and expansion pathways [143].

Due to PD-1 expression on activated T cells, physiologically activated lymphocytes can be distinguished from cells exhausted by prolonged antigen stimulation. PD-1 is referred to as a marker of lymphocyte exhaustion, but it is not an independent determinant of such cell character [142]. Lymphocytes under stable conditions also show some PD-1 expression, but this is not linked to a characteristic exhaustion phenotype [144].

PD-1, by inhibiting the activation of autoreactive lymphocytes, prevents autoimmunity and, as a negative regulator, plays a role in the immune tolerance of B and T cells [145,146].

The PD-1 molecule has two ligands: PD-L1 and PD-L2. These are widely distributed in body tissues [146]. PD-L1 expression is found in non-hematological cells, non-lymphoid organs, and even some tumors [147]. PD-L1 plays an important role in maintaining the balance between effector and regulatory T cells. PD-L1 can control the development of regulatory T cells in lymphoid organs, promote their development in target tissues, and thus protect against tissue destruction [148]. 

The PD-1/PD-L1 pathway has vital role in the etiopathogenesis of autoimmune and allergic diseases and in transplant rejection. The effects of blocking this pathway can be used to treat these diseases [146].

PD-1 is an inhibitory co-stimulator, appearing in the early activation phase of T cells after CTLA-4. It counteracts the positive signals transmitted by T cell receptor (TCR) and CD28 by engaging its PDL-1 and/or PD-2 ligands. The PD-1 pathway modulates the T cell function, tolerance, and return to immune homeostasis [149,150]. Its blockade can have profound effects on the host’s physiology. PD-1 deficiency affects the lack of inhibition of the proliferation of activated lymphocytes, whereas the high and prolonged expression of PD1 and its ligands is common during chronic inflammation and cancer development. Inhibition of the PD-1/PD-L1 pathway may improve T cell function in response to infection and tumor development [151,152].

Animals genetically lacking the Pdcd1 gene (encoding PD-1) develop accelerated autoimmunity. A study in PD-1-deficient mice resulted in spontaneously developed lupus-like proliferative arthritis and glomerulonephritis with predominant IgG3 deposition, as well as dilated cardiomyopathy with troponin I antibodies. These symptoms were more severe in those animals which additionally had a Fas protein mutation. This experiment indicated an important role for PD-1 in maintaining peripheral self-tolerance by negatively regulating the immune response [153,154].

Dmowski et al. suggested that the development of endometriosis results from an innate or adaptive defect in the immune response [155]. At the same time, women with endometriosis are more prone to autoimmune diseases such as systemic lupus erythematosus, Sjögren’s syndrome, rheumatoid arthritis, multiple sclerosis, and allergies. The association between endometriosis and systemic lupus erythematosus (SLE) was observed by Pasoto et al. who highlighted the similarity of the symptoms in endometriosis and SLE. Although none of the patients treated for endometriosis were diagnosed with SLE using the so-called minimum criteria for this disease, the patients in that study suffered significantly more often from joint pain (62%) and muscle pain (18%) compared to the control group. The frequency of these symptoms was similar to that reported by patients with SLE [63,156]. This suggests that there is a common source of these immune disorders.

It has been proven that dysfunction of the PD-1/PD-L1 pathway is involved in a much larger number of autoimmune disorders. By using a model of collagen-induced arthritis, Wang et al. demonstrated the reduced expression of pro-inflammatory cytokines by administration of a recombinant anti-PD-L1 antibody [157]. Kobayashi et al. indicated that PD-1/PD-L1 axis disorders occur in patients with Sjögren’s syndrome [158]. In mouse models, autoreactive T cells in the target organ expressed high PD-1 levels [151,152,159]. Inflammation induced during target tissue destruction can increase PD-L1 expression [150,160], providing a means to regulate autoreactive T cells. In NOD mice, PD-L1 expression in pancreatic islets is more important than PD-L1 expression in hematopoietic cells when it comes to its protective function against the development of diabetes [161]. The PD-1/PD-L1 pathway controls autoreactive T cells at multiple developmental stages: first, during the development of the T-cell repertoire; second, during the priming and differentiation of effector T cells; and finally, during the acquisition of effector functions in target organs [151].

Our previous study revealed significantly higher expression of PD-1 antigen and PD-L1 antigen among CD4+ T lymphocytes, CD8+ T lymphocytes, and CD19+ B lymphocytes. Additionally, in patients with pelvic pain syndrome and endometriosis, there was a positive correlation between CD19+PD-L1 B cells and the percentage of CD3-CD16+CD56+ NK cells, and a positive correlation between the percentage of CD8+PD-L1 T cells and the percentage of CD4+CD25+highFoxp3+ regulatory T cells. The complexity and multifactorial etiology of endometriosis suggests that negative costimulation has a role in its development [162]. 

Zamani et al. investigated the relationship of PD-1 with infertility by evaluating PD-1 gene polymorphisms at the level of single nucleotide polymorphism (SNP) of the genome, as well as susceptibility to antisperm antibody-associated infertility in an Iranian group of infertile patients. These authors demonstrated an association between impaired PD-1-related immunomodulation and the development of antisperm antibodies, confirming the presence of PD-1/PD-L1 pathway dysfunction as one of the elements in the pathogenesis of infertility [163]. 

The literature indicates an important role of the soluble form of PD-1, which promotes the T cell response by blocking the negative costimulation resulting from the activation of PD-1 membrane forms [164]. In addition, several clinical studies have analyzed the relationship between blood levels of the soluble form of PD-1 and the clinicopathological characteristics in cancer patients, demonstrating that this form of PD-1 can be used as a predictive factor [165,166].

### 5.2. Cytotoxic T-Lymphocyte Antigen-4 (CTLA-4) and Its Ligand, CD86 Antigen

The CTLA-4 molecule, like PD-1, belongs to the family of type I membrane receptors and is an important checkpoint in signalling between cells of the immune system. Its action is inhibitory and represents a negative feedback signal in the development of a specific immune response [167]. CTLA-4 expression is observed both on the surface of CD4+ T, T, regulatory T cells and on B 19+ cells subjected to antigen stimulation. Stimulated T cells express CTLA-4 in the presence of CD28. Both molecules have the ability to interact with the same ligands—B.7-1 (CD80) or B.7-2 (CD86)—but CTLA-4, according to different sources, shows a 10–50 times stronger affinity for them [168,169]. CTLA-4 promotes lymphocytes to remain in an anergic state. At the cellular level, the synthesis of cyclin D3 and cdk4/cdk6 kinases is inhibited, degradation of the p27 inhibitor protein and expression of cyclin D2 are enhanced. Cells are not activated even after recognition of a specific antigen, which results in no transition from G0 to G1 phase. The costimulatory signals that induce virgin lymphocyte expression are not sufficient in this case [170,171,172,173].

The mechanism regulating the immune response has two stages. The initial antigen-mediated lymphocyte activation with major histocompatibility complex (MHC) proteins and T cell receptor (TCR) is followed by a co-stimulatory signal. This signal is important because it can modify the further course of the process. The activating signal from the interaction between CD28 on T cells and CD80 (B7.1) and CD86 (B7.2) on antigen-presenting cells (APC) leads to lymphocyte proliferation, increased survival and differentiation through cytokine production. CTLA-4 is one of the first negative regulators to compete directly with CD28 by binding to the same ligands. Strong TCR activation is a stimulus that increases CTLA-4 expression on the lymphocyte surface. As a result, after binding to CD80/CD86, the lymphocyte enters an anergic state—IL-2 production and proliferation are inhibited [167,169,170,174]. The CTLA-4 molecule is also involved in other aspects of immune response regulation. In animal models, the genetic deficiency of CTLA-4 impaired Treg suppressor functions. The constitutive expression of CTLA-4 on Treg can sequester or internalise the CD80/CD86 receptor on APC cells, leading to reduced CD28 co-stimulation. The absence of a co-stimulatory signal may lead to reduced T cell proliferation and reduced effector function. Chinese researchers demonstrated that CTLA-4 blockade directly inhibits the autoimmune response in vitro in a mouse model of endometriosis. By using anti-CTLA-4 antibody, a gradual reduction of CD4+ CD25+ Treg cells was demonstrated and—through broken immunotolerance—inhibited proliferation and invasion of ectopic endometrial cells [172,174]. 

Studies on gene polymorphism for the CTLA-4 molecule did not show a statistically significant relationship with the occurrence of endometriosis. However, this does not exclude the molecule’s involvement in local immune response disorders in the disease environment. Experiments in a mouse model on the therapeutic potential of the anti-CTLA-4 antibody have shown that it may be an important tool to inhibit the progression of endometriosis by regulating the overproduction of CD4+ CD25+ Treg cells, a phenomenon repeatedly reported in the literature [175]. 

In our own study, significantly higher CTLA-4 antigen expression was observed among CD8+ T lymphocytes in peripheral blood. No statistically significant differences were noted between the endometriosis patients and controls in other groups of lymphocytes. At the same time, a positive correlation was reported between the stage of endometriosis and the percentage of CD4+CTLA-4+ T lymphocytes and CD8+CTLA-4 T lymphocytes. There was also a weak positive correlation between the stage of endometriosis and the percentage of CD19+CTLA-4+ B lymphocytes. These data prove to be interesting in the context of the reports of Hegel et al. who, in their studies on CTLA-4 in an animal model, demonstrated that CD8+ T cells without CTLA- 4 expression showed significantly increased production of IFN-γ and granzyme B, as well as enhanced cytolytic function. Nonetheless, CTLA-4 expression had no effect on the proliferation of these cells [176]. So far, there have been no other studies published discussing CTLA-4 expression in endometriosis. All these data further indicate the important role of immunosuppressive mechanisms in the development of the disease. Furthermore, significantly higher percentages of CD4+/CTLA-4 and CD8+/CTLA-4 T lymphocytes were observed in patients with endometriosis and intraoperative adhesions. This information is somewhat in opposition to the reports of Holsti et al. who, using a mouse model, investigated the effect of costimulatory signals on postoperative adhesion formation. According to their reports, the blockade of CD28 interaction with CD80 and CD86 ligands completely abolishes the formation of adhesions. However, the inhibitory costimulatory signal involving CTLA-4 does not significantly affect their formation [177]. A native, soluble form of CTLA-4 has been described [178]. The presence of high serum concentrations of this form has been correlated with several autoimmune diseases [179]. However, no significant changes in its concentration either in peripheral blood or in peritoneal fluid in endometriosis patients were found.

### 5.3. The CD200 Molecule and Its Receptor, CD200R

The molecules exhibiting immunomodulatory activity are CD200 together with their receptor CD200 (CD200R). CD200 is a surface protein found on cells of both myeloid and lymphoid lineages, as well as in many tissues. It interacts with the CD200 ligand [180,181]; these interactions can result in the induction of immune tolerance, regulation of cytokine release, and participation in cell maturation, adhesion and chemotaxis [180,182]. This pathway has been shown to be involved in the inhibition of excessive inflammatory responses, in the pathogenesis of cancer, and in the suppression of autoimmune responses [180,183]. 

The mechanisms of immunosuppression in the endometriotic environment are not fully explained, but the relationships demonstrated so far suggest that the regulatory molecule CD200 with its ligand is an important element involved in their induction. 

Among its many properties, CD200 increases the production of indoleamine 2,3-dioxygenase, suppresses NK cells, neutrophils and macrophages, and stimulates Treg cell growth. The probability of ectopic endometrial implantation and growth is greater in an environment of relative immune tolerance. Additionally, Treg cells are postulated to have a key role in the formation and growth of ectopic implants [184,185]. An important factor involved in the stabilisation of endometrial implants is also the inhibited activity of NK cells in their environment [186,187]. The only report to date on the role of CD200 in endometriosis indicates that the accumulation of the soluble receptor is elevated in the eutopic, secretory phase endometrium in women with endometriosis. Considering the retrograde menstruation theory, this may explain the development of ectopic lesions. The authors of the above report propose that the quantification of soluble CD200 in menstrual blood is a method to verify a predisposition to endometriosis development [188]. 

Hoek et al. found that CD200-deficient mice increased endogenous macrophage/myeloid cell activation in the central nervous system, which was associated with an increased susceptibility to encephalomyelitis and arthritis [189]. Administration of an anti-CD200R antibody to disrupt the CD200-CD200R interaction also increased the susceptibility of mice to arthritis. Furthermore, Broderick et al. discovered that blockade of CD200 caused early onset of experimental autoimmune uveitis in mice [190]. These reports indicated that CD200 may have a role in the development of endometriosis, which in many ways resembles an autoimmune disease [188]. 

The authors also argued that the expression of the CD200 and CD200R molecules is related to the continuous antigenic stimulation that occurs during chronic inflammation. In addition, it was found that the level of CD200R expression on CD4+ T cells was related to the current infection status [191]. Moreover, the immunoregulatory function of CD200 is expressed through its effect on the population, and the soluble form of CD200 expressed in plasma affects the expansion of CD4+CD25+highFoxp3+ regulatory T cells. Pallasch et al. demonstrated that blocking the CD200-CD200R interaction by anti-CD200 antibodies reduced the population of regulatory T cells [192].

Our own studies revealed a significantly higher percentage of CD4+/CD200 T lymphocytes, CD8+CD200 T lymphocytes, and CD19+CD200 B lymphocytes. At the same time, the expression of CD200R antigens on the surface of the same lymphocyte populations was significantly lower compared to the control group. A strong positive correlation was observed between the stage of endometriosis and the percentage of CD8+CD200 T lymphocytes and CD8+CD200R T lymphocytes. In patients with endometriosis treated for infertility, a negative correlation was noted between the percentage of CD3+CD16+CD56+ NK cells and CD8+CD200 T cells, CD19+CD200 B cells and CD8+CD200R T cells. In patients with endometriosis and adhesion disease, there was reported a positive correlation between the percentages of CD4+CD25+highFoxp3+ regulatory T cells and CD19+CD200 cells, as well as a negative correlation between the percentages of CD3+CD16+CD56+ NK cells and 19+CD200 B cells, which indicated a particularly intense immunosuppression in this group. 

Additionally, our study showed that there was a statistically lower concentration of the soluble form of CD200 in the peripheral blood plasma in the group of women with endometriosis compared to the control group. The study also showed that this form of CD200 was quantifiable in the peritoneal fluid. A similar study on the soluble form of CD200 was carried out by Gorczynski et al. They showed that tumor growth in vivo can be monitored by levels of soluble CD200 in the serum of animals with tumors [193]. In contrast, Moreaux et al. found significant overexpression of CD200 in various tumors, compared to normal cells or tissues [194]. Their study suggested that CD200 might be a potential therapeutic target and prognostic factor, both in cancer and endometriosis.

## 6. Conclusions

It is estimated that up to 10% of women of reproductive age have endometriosis. This disease is characterized by chronic pelvic pain, is associated with infertility, and is a factor in the development of diseases such as asthma, cardiovascular disease, ovarian and breast cancer, melanoma, and rheumatoid arthritis. 

Endometriosis is increasingly discussed as an autoimmune disease. It turns out that immune cells, such as neutrophils, macrophages, NK cells and dendritic cells, may play a special role in the angiogenesis, growth, and invasion of endometriotic cells. Immune cells secrete cytokines and defensins, which also affect the endometriosis environment. Immune checkpoint inhibitors should be responsible for controlling the immune response, but in patients with endometriosis their levels are observed to differ from those in healthy patients. 

There is still little research on the immune system and endometriosis. An in-depth study is needed to determine the exact mechanisms of disease pathogenesis, which will help us to understand the mechanisms and, in the future, may allow the use of T-cell-targeted immunotherapy or the administration of immune checkpoint inhibitors as a promising method of treatment for patients with endometriosis.

## Figures and Tables

**Figure 1 cells-11-02028-f001:**
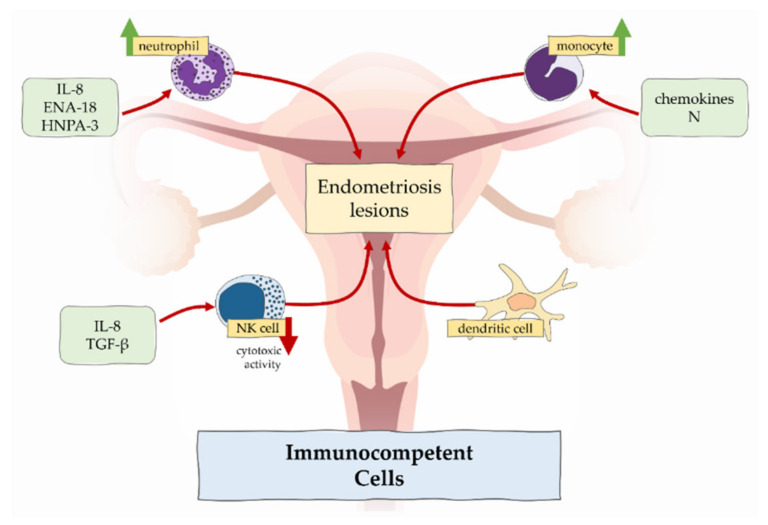
The role of immunocompetent cells in endometriosis.

**Figure 2 cells-11-02028-f002:**
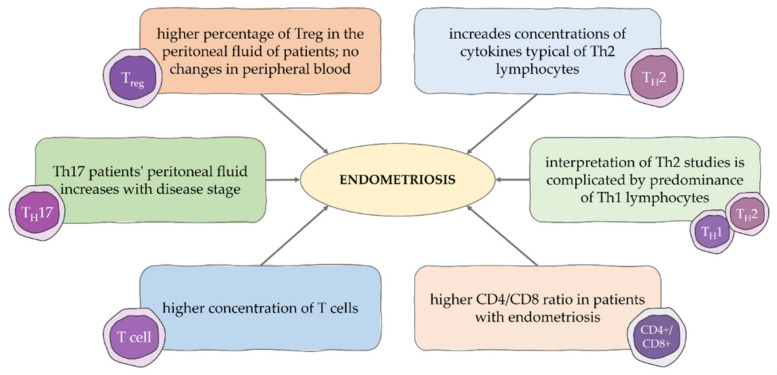
Participation of T cell subpopulation in the pathogenesis of endometriosis.

**Table 1 cells-11-02028-t001:** The potential role of selected cytokines in the pathogenesis of endometriosis.

Cytokine	Potential Role
INF-γ	interacts with IL-2 in regulating the balance between Th1 and Th2 cells; conflicting research results
IL-1	regulation of immune and inflammatory responses
IL-2	role in the pathogenesis of endometriosis is not clearly defined
IL-6	increase in haptoglobin, which “protects” endometrial implants from immune surveillance by reducing phagocytosis
IL-8 and TNF-α	are factors that facilitate the adhesion of endometrial cells to the peritoneum and may contribute to the progression of the disease
IL-4 and IL-10	higher values of IL-4 and IL-10 in the peritoneal fluid confirms that the development of endometriosis is accompanied by the activation of a Th2-type immune response
TGF-β	may be responsible for inhibiting NK cell activity in the peritoneal fluid
VEGF	potent heparin-binding angiogenic factor

## Data Availability

Not applicable.

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
