# Peer review of "The Role of the Immune System in the Development of Endometriosis"

_cells, 2022, doi:10.3390/cells11132028_

Round 1

Reviewer 1 Report

The paper deals with a very interesting and important research topic from the point of view of modern gynecology. The work is a very valuable study on the current state of knowledge on the role of the immune system in the formation and development of endometriosis.

However, I believe that it could be supplemented with the indicated issues that will enrich the substantive value of the work, as well as figures and a table that will facilitate the analysis of the issues presented.

1. In the introduction, please provide the classifications of endometriosis, such as ASRM or ENZIAN.

2. Although endometriosis is still called the "disease of a thousand hypotheses" and there are many theories trying to explain the emergence and development of this disease, and none of them clearly explains the survival and proliferation mechanism of the ectopic endometrium beyond its natural place of occurrence, I believe that the immunological theory of prof. Dmowski is worth mentioning.

3. The chapter "The role of selected cytokines in the pathogenesis of endometriosis" should also be supplemented with the role of the IL-1 family cytokine network, which is important in the pathogenesis of endometriosis.

4. The authors presented the role of immunocompetent cells in the development of endometriosis in a very interesting way, which may also be illustrated in the figure, which will increase the value of the work and make it easier for the reader to analyze.

5. The roles of the T cell subpopulation in the pathogenesis of endometriosis may be presented in a diagram which will be clearer.

6. The role of selected cytokines in the pathogenesis of endometriosis is noteworthy, but maybe the data should be summarized in a table.

Author Response

Dear Reviewer,

Thank you for the opportunity to correct our paper, entitled: The role of immune system in the development of endometriosis. We have thoroughly revised the text and implemented the corrections suggested by you. Please find the point-by-point answers to your concerns:

Reviewer 1

The paper deals with a very interesting and important research topic from the point of view of modern gynecology. The work is a very valuable study on the current state of knowledge on the role of the immune system in the formation and development of endometriosis.

However, I believe that it could be supplemented with the indicated issues that will enrich the substantive value of the work, as well as figures and a table that will facilitate the analysis of the issues presented.

  1. In the introduction, please provide the classifications of endometriosis, such as ASRM or ENZIAN.

RE: In the introduction, we added the revised American Society for Reproductive Medicine (rASRM) classification, the ENZIAN classification, the endometriosis fertility index (EFI), and the American Association of Gynecological Laparoscopists (AAGL) and described the differences between them.

  1. Although endometriosis is still called the "disease of a thousand hypotheses" and there are many theories trying to explain the emergence and development of this disease, and none of them clearly explains the survival and proliferation mechanism of the ectopic endometrium beyond its natural place of occurrence, I believe that the immunological theory of prof. Dmowski is worth mentioning.

RE: Also in the introduction, we added a paragraph referring to Professor Dmowski's immunological theory.

  1. The chapter "The role of selected cytokines in the pathogenesis of endometriosis" should also be supplemented with the role of the IL-1 family cytokine network, which is important in the pathogenesis of endometriosis.

RE: We have included an additional section relating to the role of the IL-1 family in endometriosis.

  1. The authors presented the role of immunocompetent cells in the development of endometriosis in a very interesting way, which may also be illustrated in the figure, which will increase the value of the work and make it easier for the reader to analyze.

RE: Thank you for your opinion, the role of immunocompetent cells in the pathogenesis of endometriosis in the form of a figure was added to Chapter 2.

  1. The roles of the T cell subpopulation in the pathogenesis of endometriosis may be presented in a diagram which will be clearer.

RE: We added diagram showing participation of the T cell subpopulation in pathogenesis of endometriosis in chapter 2.

  1. The role of selected cytokines in the pathogenesis of endometriosis is noteworthy, but maybe the data should be summarized in a table.

RE: A table summarizing the role of cytokines in the pathogenesis of endometriosis has been added to Chapter 3.

Moreover, we enclose the authorship change form, as we decided to add one author (Rafał Hrynkiewicz), whose role was crucial to create the figures.

We do hope, that in the current form the paper can be accepted.

On behalf of the Authors,

Paulina Niedźwiedzka-Rystwej

Reviewer 2 Report

The reviewer thinks the manuscript is well-reviewed in terms of the involvement of immune response, especially immune checkpoint inhibitors,  in the pathogenesis of endometriosis. After correcting some minor grammatical errors, the reviewer believes the manuscript is acceptable in its current form.   

Minor comments

The manuscript includes several grammatical errors. Some examples are shown below. Please check those throughout the manuscript. 

#1. Line 50: Is "if" "of"? 

#2. Line 100: Is "CD56^bright CD16" "CD56^bright CD16-"?

#3. Line 109: Is "(95-98)" reference the number?

#4. Lines 210-212: Please check if the sentence is correct.

Author Response

Dear Reviewer,

Thank you for the opportunity to correct our paper, entitled: The role of immune system in the development of endometriosis. We have thoroughly revised the text and implemented the corrections suggested by you. Please find the point-by-point answers to your concerns:

Reviewer 2

The reviewer thinks the manuscript is well-reviewed in terms of the involvement of immune response, especially immune checkpoint inhibitors,  in the pathogenesis of endometriosis. After correcting some minor grammatical errors, the reviewer believes the manuscript is acceptable in its current form.   

Minor comments

The manuscript includes several grammatical errors. Some examples are shown below. Please check those throughout the manuscript. 

 #1. Line 50: Is "if" "of"? 

#2. Line 100: Is "CD56^bright CD16" "CD56^bright CD16-"?

 #3. Line 109: Is "(95-98)" reference the number?

 #4. Lines 210-212: Please check if the sentence is correct.

RE: Thank you for your positive review of our work. Minor errors that were pointed out have been corrected.

Moreover, we enclose the authorship change form, as we decided to add one author (Rafał Hrynkiewicz), whose role was crucial to create the figures asked by one of the Reviewer.

We do hope, that in the current form the paper can be accepted.

On behalf of the Authors,

Paulina Niedźwiedzka-Rystwej